# Kalman Filter, Sensor Fusion, and Constrained Regression: Equivalences and Insights

**Maria Jahja**
Department of Statistics
Carnegie Mellon University
Pittsburgh, PA 15213
maria@stat.cmu.edu

**David Farrow**
Computational Biology Department
Carnegie Mellon University
Pittsburgh, PA 15213
dfarrow0@gmail.com

**Roni Rosenfeld**
Machine Learning Department
Carnegie Mellon University
Pittsburgh, PA 15213
roni@cs.cmu.edu

**Ryan J. Tibshirani**
Department of Statistics
Machine Learning Department
Carnegie Mellon University
Pittsburgh, PA 15213
ryantibs@stat.cmu.edu

## Abstract

The Kalman filter (KF) is one of the most widely used tools for data assimilation and sequential estimation. In this work, we show that the state estimates from the KF in a standard linear dynamical system setting are equivalent to those given by the KF in a transformed system, with infinite process noise (i.e., a "flat prior") and an augmented measurement space. This reformulation—which we refer to as augmented measurement sensor fusion (SF)—is conceptually interesting, because the transformed system here is seemingly static (as there is effectively no process model), but we can still capture the state dynamics inherent to the KF by folding the process model into the measurement space. Further, this reformulation of the KF turns out to be useful in settings in which past states are observed eventually (at some lag). Here, when the measurement noise covariance is estimated by the empirical covariance, we show that the state predictions from SF are equivalent to those from a regression of past states on past measurements, subject to particular linear constraints (reflecting the relationships encoded in the measurement map). This allows us to port standard ideas (say, regularization methods) in regression over to dynamical systems. For example, we can posit multiple candidate process models, fold all of them into the measurement model, transform to the regression perspective, and apply $\ell_1$ penalization to perform process model selection. We give various empirical demonstrations, and focus on an application to nowcasting the weekly incidence of influenza in the US.

## 1 Introduction

Let $x_t \in \mathbb{R}^k$, $t = 1, 2, 3, \ldots$ denote states and $z_t \in \mathbb{R}^d$, $t = 1, 2, 3, \ldots$ denote measurements evolving according to the time-invariant linear dynamical system:

$$x_t = F x_{t-1} + \delta_t, \tag{1}$$
$$z_t = H x_t + \epsilon_t, \tag{2}$$

for $t = 1, 2, 3, \ldots$. We assume the noise terms $\delta_t, \epsilon_t$ have mean zero and covariances $Q \in \mathbb{R}^{k \times k}$ and $R \in \mathbb{R}^{d \times d}$, respectively, for all $t = 1, 2, 3, \ldots$. Also, we assume that the initial state $x_0$ and all noise terms are mutually independent. We call (1) the process model and (2) the measurement model.

**Kalman filter.** The Kalman filter (KF) [Kalman, 1960] is a method for sequential estimation in the model (1), (2). Given past estimates $\hat{x}_1, \ldots, \hat{x}_t$ and measurements $z_1, \ldots, z_{t+1}$, we form an estimate $\hat{x}_{t+1}$ of the state $x_{t+1}$ via

$$\bar{x}_{t+1} = F\hat{x}_t, \tag{3}$$

$$\hat{x}_{t+1} = \bar{x}_{t+1} + K_{t+1}(z_{t+1} - H\bar{x}_{t+1}), \tag{4}$$

where $K_{t+1} \in \mathbb{R}^{k \times d}$ is called the *Kalman gain* (at time $t + 1$). It is itself updated sequentially, via

$$\bar{P}_{t+1} = FP_tF^T + Q, \tag{5}$$

$$K_{t+1} = \bar{P}_{t+1}H^T(H\bar{P}_{t+1}H^T + R)^{-1}, \tag{6}$$

$$P_{t+1} = (I - K_{t+1}H)\bar{P}_{t+1}. \tag{7}$$

where $P_{t+1} \in \mathbb{R}^{k \times k}$ denotes the state error covariance (at time $t + 1$). The step (3) is often called the *predict* step: we form an intermediate estimate $\bar{x}_{t+1}$ of the state based on the process model and our estimate at the previous time point. The step (4) is often called the *update* step: we update our estimate $\hat{x}_{t+1}$ based on the measurement model and the measurement $z_{t+1}$.

Under the data model (1), (2) and the conditions on the noise stated above, the Kalman filter attains the optimal mean squared error $\mathbb{E}\|\hat{x}_t - x_t\|_2^2$ among all linear unbiased filters, at each $t = 1, 2, 3, \ldots$. When the initial state $x_0$ and all noise terms are Gaussian, the Kalman filter estimates exactly reduce to the Bayes estimates $\hat{x}_t = \mathbb{E}(x_t|z_1, \ldots, z_t)$, $t = 1, 2, 3, \ldots$. Numerous important extensions have been proposed, e.g., the ensemble Kalman filter (EnKF) [Evensen, 1994, Houtekamer and Mitchell, 1998], which approximates the noise process covariance $Q$ by a sample covariance in an ensemble of state predictions, as well as the extended Kalman filter (EKF) [Smith et al., 1962] and unscented Kalman filter (UKF) [Julier and Uhlmann, 1997], which both allow for nonlinearities in the process model. Particle filtering (PF) [Gordon et al., 1993] has more recently become a popular approach for modeling complex dynamics. PF adaptively approximates the posterior distribution, and in doing so, avoids the linear and Gaussian assumptions inherent to the KF. This flexibility comes at the cost of a greater computational burden.

In this paper, we revisit the standard KF (3), (4) and show that its estimates $\hat{x}_{t+1}$, $t = 0, 1, 2, \ldots$ are equivalent to those from the KF applied to a transformed system, with infinite process noise and an augmented measurement space. At first glance, this is perhaps surprising, because the transformed system effectively lacks a process model and is therefore seemingly static; however, it is able to take the state dynamics into account as part of its measurement model. Importantly, this reformulation of the KF leads us to derive a second, key reformulation for problems in which past states are observed (at some lag). This second reformulation is the methodological crux of our paper: it is a constrained regression approach for predicting states from measurements, motivated by (derived from) SF and the KF. We illustrate its effectiveness in an application to nowcasting weekly influenza levels in the US.

**Sensor fusion.** If we let the noise covariance in the process model diverge to infinity, $Q \to \infty$[1], then the Kalman filter estimate in (3), (4) simplifies to

$$\hat{x}_{t+1} = (H^TR^{-1}H)^{-1}H^TR^{-1}z_{t+1}. \tag{8}$$

This can be verified by rewriting the Kalman gain as $K_{t+1} = (\bar{P}_{t+1}^{-1} + H^TR^{-1}H)^{-1}H^TR^{-1}$, and observing that $\bar{P}_{t+1}^{-1} \to 0$ as $Q \to \infty$. Alternatively, we can verify this by specializing to the case of Gaussian noise: as $\text{tr}(Q) \to \infty$, we approach a flat prior, and the Kalman filter (Bayes estimator) just maximizes the likelihood of $z_{t+1}|x_{t+1}$. From the measurement model (2) (assuming Gaussian noise), this is a weighted regression of $z_{t+1}$ on the measurement map $H$, precisely as in (8).

We will call (8) the *sensor fusion* (SF) estimate (at time $t + 1$).[2] In this setting, we will also refer to the measurements as *sensors*. As defined, sensor fusion is a special case of the Kalman filter when there is infinite process noise; said differently, it is a special case of the Kalman filter when there is no process model at all. Thus, looking at (8), the state dynamics have apparently been completely lost. Perhaps surprisingly, as we will show shortly, these dynamics can be exactly recovered by augmenting the measurement vector $z_{t+1}$ with the KF intermediate prediction $\bar{x}_{t+1} = F\hat{x}_t$ in (3) (and adjusting the map $H$ and covariance $R$ appropriately). We summarize this and our other contributions next.

**Summary of contributions.** An outline of our contributions in this paper is as follows.

1. We show in Section 2 that, if we take the KF intermediate prediction $\bar{x}_{t+1}$ in (3), append it to the measurement vector $z_{t+1}$, and perform SF (8) (with an appropriately adjusted $H, R$), then the result is exactly the KF estimate (4).

2. We show in Section 3 that, if we are in a problem setting in which past states are observed (at some lag, which is the case in the flu nowcasting application), and we replace the noise covariance $R$ from the measurement model by the empirical covariance on past data, then the sensor fusion estimate (8) can be written as $\hat{B}^T z_{t+1}$, where $\hat{B} \in \mathbb{R}^{d \times k}$ is a matrix of coefficients that solves a regression problem of the states on the measurements (using past data), subject to the equality constraint $H^T \hat{B} = I$.

3. We demonstrate the effectiveness of our new regression formulation of SF in Section 4 by describing an application of this methodology to nowcasting the incidence of weekly flu in the US. This achieves state-of-the art performance in this problem.

4. We present in Section 5 some extensions of the regression formulation of SF; they do not have direct equivalences to SF (or the KF), but are intuitive and extend dynamical systems modeling in new directions (e.g., using $\ell_1$ penalization to perform a kind of process model selection).

We make several remarks. The equivalences described in points 1–3 above are deterministic (they do not require the modeling assumptions (1), (2), or any modeling assumptions whatsoever). Further, even though their proofs are elementary (they are purely linear algebraic) and the setting is a classical one (linear dynamical systems), these equivalences are—as far as we can tell—new results. They deserve to be widely known and may have implications beyond what is explored in this paper.

For example, the regression formulation of SF may still be a useful perspective for problems in which past states are fully unobserved (this being the case in most KF applications). In such problems, we may consider using *smoothed* estimates of past states, obtained by running a backward version of the KF forward recursions (3)–(7) (see, e.g., Chapter 7 of Anderson and Moore [1979]), for the purposes of the regression formulation. As another example, the SF view of the KF may be a useful formulation for the purposes of estimating the covariances $R, Q$, or the maps $F, H$, or all of them; in this paper, we assume that $F, H, R, Q$ are known (except for in the regression formulation of SF, in which $R$ is unknown but past states are available); in general, there are well-developed methods for estimating $F, H, R, Q$ such as *subspace identification* algorithms (see, e.g., Overshee and Moor [1996]), and it may be interesting to see if the SF perspective offers any advantages here.

**Related work.** The Kalman filter and its extensions, as previously referenced (EnKF, EKF, UKF), are the de facto standard in state estimation and tracking problems; the literature surrounding them is enormous and we cannot give a thorough treatment. Various authors have pointed out the simple fact that maximum likelihood estimate in (8), which we call sensor fusion, is the limit of the KF as the noise covariance in the process model approaches infinity (see, e.g., Chapter 5.9 of Brown and Hwang [2012]). We have not, however, seen any authors note that this static model can recover the KF by augmenting the measurement vector with the KF intermediate prediction (Theorem 1).

Along the lines of our second equivalence (Theorem 2), there is older work in the statistical calibration literature that studies the relationships between the regressions of $y$ on $x$ and $x$ on $y$ (for multivariate $x, y$, see Brown [1982]). This is somewhat related to our result, since we show that a *backwards* or *indirect* approach, which models $z_{t+1} | x_{t+1}$, is actually equivalent to a *forwards* or *direct* approach, which predicts $x_{t+1}$ from $z_{t+1}$ via regression. However, the details are quite different.

Finally, our SF methodology in the flu nowcasting application blends together individual predictors in a way that resembles *linear stacking* [Wolpert, 1992, Breiman, 1996]. In fact, one implication of our choice of measurement map $H$ in the flu nowcasting problem, as well as the constraints in our regression formulation of SF, is that all regression weights must sum to 1, which is the standard in linear stacking as well. However, the equality constraints in our regression formulation are quite a bit more complex, and reflect aspects of the sensor hierarchy that linear stacking would not.

## 2   Equivalence between KF and SF

As already discussed, the sensor fusion estimate (8) is a limiting case of the Kalman filter (3), (4), and initially, it seems, one rather limited in scope: there is effectively no process model (as we have sent the process variance to infinity). However, as we show next, the KF is actually itself a special case of SF, when we augment the measurement vector by the KF intermediate predictions, and appropriately adjust the measurement map $H$ and noise covariance $R$. The proof is elementary, a consequence of the Woodbury matrix and related manipulations. It is given in the supplement.

**Theorem 1.** *At each time $t = 0, 1, 2, \ldots$, suppose we augment our measurement vector by defining $\tilde{z}_{t+1} = (z_{t+1}, \bar{x}_{t+1}) \in \mathbb{R}^{d+k}$, where $\bar{x}_{t+1} = F\hat{x}_t$ is the KF intermediate prediction at time $t + 1$. Suppose that we also augment our measurement map by defining $\tilde{H} \in \mathbb{R}^{(d+k) \times k}$ to be the rowwise concatenation of $H$ and the identity matrix $I \in \mathbb{R}^{k \times k}$. Furthermore, suppose we define an augmented measurement noise covariance*

$$\tilde{R}_{t+1} = \begin{bmatrix} R & 0 \\ 0 & \bar{P}_{t+1} \end{bmatrix}, \tag{9}$$

*where $\bar{P}_{t+1}$ is the KF intermediate error covariance at time $t + 1$ (as in (5)). Then applying SF to the augmented system produces an estimate at $t + 1$ that equals the KF estimate,*

$$(\tilde{H}^T \tilde{R}_{t+1}^{-1} \tilde{H})^{-1} \tilde{H}^T \tilde{R}_{t+1}^{-1} \tilde{z}_{t+1} = \bar{x}_{t+1} + K_{t+1}(z_{t+1} - H\bar{x}_{t+1}), \tag{10}$$

*where $K_{t+1}$ is the Kalman gain at $t + 1$ (as in (6)).*

**Remark 1.** We can think of the last state estimate $\hat{x}_t$ in the theorem (which is propagated forward via $\bar{x}_{t+1} = F\hat{x}_t$) as the previous output from SF itself, when applied to the appropriate augmented system. More precisely, by induction, Theorem 1 says that iteratively applying SF to $\tilde{z}_{t+1}, \tilde{H}, \tilde{R}_{t+1}$ across times $t = 0, 1, 2, \ldots$, where each $\bar{x}_{t+1} = F\hat{x}_t$ is the intermediate prediction using the last SF estimate $\hat{x}_t$, produces a sequence $\hat{x}_{t+1}$, $t = 0, 1, 2, \ldots$ that matches the state estimates from the KF.

**Remark 2.** The result in Theorem 1 can be seen from a Bayesian perspective, as was pointed out by an anonymous reviewer. When the initial state $x_0$ and all noise terms in (1), (2) are Gaussian, recall the KF reduces to the Bayes estimator. Here the posterior is the product of a Gaussian likelihood and Gaussian prior, and is thus itself Gaussian. (The proof of this standard fact uses similar arguments to the proof of Theorem 1.) Meanwhile, in augmented SF, we can view the Gaussian likelihood being maximized as the product of the Gaussian density of $z_{t+1}$ and that of $\bar{x}_{t+1}$. This matches the posterior used by the KF, where the density of $\bar{x}_{t+1}$ plays the role of the prior in the KF. Therefore in each case, we are defining our estimate to be the mean of the same Gaussian distribution.

**Remark 3.** The equivalence between SF and KF can be extended beyond the case of linear process and linear measurement models. Given a nonlinear process map $f$ and a nonlinear process model $h$, suppose we define $\bar{x}_{t+1} = f(\hat{x}_t)$, $F_{t+1} = Df(\hat{x}_t)$ (the Jacobian of $f$ at $\hat{x}_t$), and $H_{t+1} = Dh(\bar{x}_{t+1})$ (the Jacobian of $h$ at $\bar{x}_{t+1}$). Suppose we define the augmented measurement vector as

$$\tilde{z}_{t+1} = \big(z_{t+1} + H_{t+1}\bar{x}_{t+1} - h(\bar{x}_{t+1}), \bar{x}_{t+1}\big), \tag{11}$$

where we have offset the measurement $z_{t+1}$ by the residual $H_{t+1}\bar{x}_{t+1} - h(\bar{x}_{t+1})$ from linearization. Suppose, as in the theorem, we define the augmented measurement map $\tilde{H}_{t+1} \in \mathbb{R}^{(d+k) \times k}$ to be the rowwise concatenation of $H_{t+1}$ and $I \in \mathbb{R}^{k \times k}$, and define $\tilde{R}_{t+1} \in \mathbb{R}^{(d+k) \times (d+k)}$ as in (9), for $\bar{P}_{t+1}$ as in (5), but with $F_{t+1}, H_{t+1}$ in place of $F, H$. In the supplement, we prove that

$$(\tilde{H}_{t+1}^T \tilde{R}_{t+1}^{-1} \tilde{H}_{t+1})^{-1} \tilde{H}_{t+1}^T \tilde{R}_{t+1}^{-1} \tilde{z}_{t+1} = \bar{x}_{t+1} + K_{t+1}\big(z_{t+1} - h(\bar{x}_{t+1})\big), \tag{12}$$

where $K_{t+1}$ is as in (6), but with $F_{t+1}, H_{t+1}$ in place of $F, H$. The right-hand side above is precisely the *extended* KF (EKF). The left-hand side is what we might call *extended* SF (ESF).

## 3   Equivalence between SF and regression

Suppose that in our linear dynamical system, at each time $t$, we observe the measurement $z_t$, make a prediction $\hat{x}_t$ for $x_t$, then later observe the state $x_t$ itself. (This setup indeed describes the influenza nowcasting problem, a central motivating example that we will describe shortly.) In such problems, we can estimate $R$ using the empirical covariance on past data. When we plug this into (8), it turns out SF reduces to a prediction from a constrained regression of past states on past measurements.

### 3.1 Equivalent regression problem

In making a prediction at time $t + 1$, we assume in this section that we observe past states. We may assume without a loss of generality that we observe the full past $x_i$, $i = 1, \ldots, t$ (if this is not the case, and we observe only some subset of the past, then the only changes to make in what follows are notational). Assuming the measurement noise covariance $R$ is unknown, we may use

$$\hat{R}_{t+1} = \frac{1}{t} \sum_{i=1}^{t} (z_i - Hx_i)(z_i - Hx_i)^T, \tag{13}$$

the empirical (uncentered) covariance based on past data, as an estimate. Under this choice, it turns out that sensor fusion (8) is exactly equivalent to a regression of states on measurements, subject to certain equality constraints. The proof is elementary, but requires detailed arguments. It is deferred until the supplement.

**Theorem 2.** *Let $\hat{R}_{t+1}$ be as in* (13) *(assumed to be invertible). Consider the SF prediction at time $t + 1$, with $\hat{R}_{t+1}$ in place of R. Denote this by $\hat{x}_{t+1} = \hat{B}^T z_{t+1}$, where*

$$\hat{B}^T = (H^T \hat{R}_{t+1}^{-1} H)^{-1} H^T \hat{R}_{t+1}^{-1}$$

*(and $H^T \hat{R}_{t+1}^{-1} H$ is assumed invertible). Each column of $\hat{B}$, denoted $\hat{b}_j \in \mathbb{R}^d$, $j = 1, \ldots, k$, solves*

$$\begin{aligned} \underset{b_j \in \mathbb{R}^d}{\text{minimize}} \quad & \sum_{i=1}^{t} (x_{ij} - b_j^T z_i)^2 \\ \text{subject to} \quad & H^T b_j = e_j, \end{aligned} \tag{14}$$

*where $e_j \in \mathbb{R}^d$ is the jth standard basis vector (all 0s except for a 1 in the jth component).*

**Remark 4.** As discussed in the introduction, the interpretation of $(H^T \hat{R}_{t+1}^{-1} H)^{-1} H^T \hat{R}_{t+1}^{-1} z_{t+1}$ as the coefficients from regressing $z_{t+1}$ (the response) onto $H$ (the covariates) is more or less immediate. Interpreting the same quantity as $\hat{B}^T z_{t+1} = (\hat{b}_1^T z_{t+1}, \ldots, \hat{b}_k^T z_{t+1})$, the predictions from historically regressing $x_i$, $i = 1, \ldots, t$ (the response) onto $z_i$, $i = 1, \ldots, t$ (the covariates), however, is much less obvious. The latter is a *forwards* or *direct* regression approach to predicting $x_{t+1}$, whereas SF was originally defined via the *backwards* or *indirect* perspective inherent to the measurement model (2).

### 3.2 Influenza nowcasting

An example that we will revisit frequently, for the rest of the paper, is the following influenza (or flu) nowcasting problem. The state variable of interest is the weekly percentage of weighted influenza-like illness (wILI), a measure of flu incidence provided by the Centers for Disease Control and Prevention (CDC), in each of the $k = 51$ US states (including DC). Because it takes time for the CDC to collect and compile this data, they release wILI values with a 1 week delay. Meanwhile, various proxies for the flu (i.e., data sources that are potentially correlated with flu incidence) are available in real time, e.g., web search volume for flu-related terms, site traffic metrics for flu-related pages, pharmaceutical sales for flu-related products, etc. We can hence train (using historical data) sensors to predict wILI, one from each data source, and plug them into sensor fusion (8) in order to "nowcast" the current flu incidence (that would otherwise remain unknown for another week).

Such a sensor fusion system for flu nowcasting, using $d = 308$ sensors (flu proxies), is described in Chapter 4 of Farrow [2016][3]. In addition to the surveillance sensors described above (search volume for flu terms, site traffic metrics for flu pages, etc.), the measurement vector in this nowcasting system also uses a sensor that is trained to make predictions of wILI using a seasonal autoregression with 3 lags (SAR3). From the KF-SF equivalence established in Section 2, we can think of this SAR3 sensor as serving the role of something like a process model, in the underlying dynamical system.

While wILI itself is available at the US state level, the data source used to train each sensor may only be available at coarser geographic resolution. Thus, importantly, each sensor outputs a prediction at a different geographic resolution (which reflects the resolution of its corresponding data source). As an

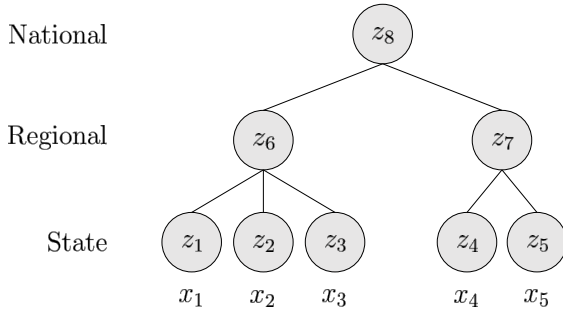

National     $z_8$

Regional     $z_6$     $z_7$

State     $z_1$ $z_2$ $z_3$    $z_4$ $z_5$

$x_1$   $x_2$   $x_3$     $x_4$   $x_5$

Figure 1: *Simplified version of the flu nowcasting problem, with $k = 5$ states and $d = 8$ sensors. We have a 3-level hierarchy, where $x_1, x_2, x_3$ are part of the first region and $x_4, x_5$ are part of the second. The national level is at the root. As for the sensors, we have one at each state, one at each region, and one at the national level. Assuming all states have equal populations, the sensor map $H$ is*

$$H = \begin{bmatrix} 1 & 0 & 0 & 0 & 0 \\ 0 & 1 & 0 & 0 & 0 \\ 0 & 0 & 1 & 0 & 0 \\ 0 & 0 & 0 & 1 & 0 \\ 0 & 0 & 0 & 0 & 1 \\ {}^1\!/_3 & {}^1\!/_3 & {}^1\!/_3 & 0 & 0 \\ 0 & 0 & 0 & {}^1\!/_2 & {}^1\!/_2 \\ {}^1\!/_5 & {}^1\!/_5 & {}^1\!/_5 & {}^1\!/_5 & {}^1\!/_5 \end{bmatrix}.$$

example, the number of visits to flu-related CDC pages are available for each US state separately; so for each US state, we train a separate sensor to predict wILI from CDC site traffic. However, counts for Wikipedia page visits are only available nationally; so we train just one sensor to predict national wILI from Wikipedia page visits.

Assuming unbiasedness of all the sensors, we construct the map $H$ in (2) so that its rows reflect the geography of the sensors. For example, if a sensor is trained on data that is available at the $i$th US state, then its associated row in $H$ is

$$(0, \ldots \underset{\underset{i}{\uparrow}}{1}, \ldots 0);$$

and if a sensor is trained on data from the aggregate of the first 3 US states, then its associated row is

$$(w_1, w_2, w_3, 0, \ldots 0),$$

for weights $w_1, w_2, w_3 > 0$ such that $w_1 + w_2 + w_3 = 1$, based on relative state populations; and so on. Figure 1 illustrates the setup in a simple example.

### 3.3 Interpreting the constraints

At a high-level, the constraints in (14) encode information about the measurement model (2). They also provide some kind of implicit regularization. Interestingly, as we will see later in Section 4, this can still be useful when used in addition to more typical (explicit) regularization.

How can we interpret these constraints? We give three interpretations, the first one specific to the flu forecasting setting, and the next two general.

**Flu interpretation.** In the flu nowcasting problem, recall, the map $H$ has rows that sum to 1, and they reflect the geographic level at which the corresponding sensors were trained (see Section 3.2). The constraints $H^T b_j = e_j$, $j = 1, \ldots, k$ can be seen in this case as a mechanism that accounts for the geographical hierachy underlying the sensors. As a concrete example, consider the simplified setup in Figure 1, and $j = 3$. The constraint $H^T b_3 = e_3$ reads:

$$b_{31} + {}^1\!/_3\, b_{36} + {}^1\!/_5\, b_{38} = 0,$$
$$b_{32} + {}^1\!/_3\, b_{36} + {}^1\!/_5\, b_{38} = 0,$$
$$b_{33} + {}^1\!/_3\, b_{36} + {}^1\!/_5\, b_{38} = 1,$$
$$b_{34} + {}^1\!/_3\, b_{37} + {}^1\!/_5\, b_{38} = 0,$$
$$b_{35} + {}^1\!/_3\, b_{37} + {}^1\!/_5\, b_{38} = 0.$$

The third line can be interpreted as follows: an increase of 1 unit in sensor $z_3$, $1/3$ units in $z_6$, and $1/5$ units in $z_8$, holding all other sensors fixed, should lead to an increase in 1 unit of our prediction for $x_3$. This is a natural consequence of the hierarchy in the sensor model (2), visualized in Figure 1. The first line can be read as: an increase of 1 unit in sensor $z_1$, $1/3$ units in $z_6$, and $1/5$ in $z_8$, with all others fixed, should not change our prediction for $x_3$. This is also natural, following from the hierachy (i.e., such a change must have been propogated by $x_1$). The other lines are similar.

**Invariance interpretation.** The SF prediction (at time $t + 1$) is $\hat{x}_{t+1} = \hat{B}^T z_{t+1}$. To denoise (i.e., estimate the mean of) the measurement $z_{t+1}$, based on the model (2), we could use $\hat{z}_{t+1} = H\hat{x}_{t+1}$. Given the denoised $\hat{z}_{t+1}$, we could then refit our state prediction via $\tilde{x}_{t+1} = \hat{B}^T \hat{z}_{t+1}$. But due to the constraint $H^T \hat{B} = I$ (a compact way of expressing $H^T \hat{b}_j = e_j$, for $j = 1, \ldots, k$), it holds that $\tilde{x}_{t+1} = \hat{B}^T H\hat{x}_{t+1} = \hat{x}_{t+1}$. This is a kind of *invariance* property. In other words, we can go from estimating states, to refitting measurements, to refitting states, etc., and in this process, our state estimates will not change.

**Generative interpretation.** Assume $t \geq k$, and fix an arbitrary $j = 1, \ldots, k$ as well as $b_j \in \mathbb{R}^k$. The constraint $H^T b_j = e_j$ implies, by taking an inner product on both sides with $x_i$, $i = 1, \ldots, k$,

$$(Hx_i)^T b_j = x_{ij}, \quad i = 1, \ldots, k.$$

If we assume $x_i$, $i = 1, \ldots, k$ are linearly independent, then the above linear equalities are not only implied by $H^T b_j = e_j$, they are actually equivalent to it. Invoking the model (2), we may rewrite the constraint $H^T b_j = e_j$ as

$$\mathbb{E}(b_j^T z_i | x_i) = x_{ij}, \quad i = 1, \ldots, k. \tag{15}$$

In the context of problem (14), this is a statement about a *generative* model for the data (as $z_i | x_i$ describes the distribution of the covariates conditional on the response). The representation in (15) shows that (14) constrains the regression estimator to have the correct conditional predictions, on average, on the data we have already seen $(x_i, z_i)$, $i = 1, \ldots, k$. (Note here we did not have to use the first $k$ time points; any past $k$ time points would suffice.)

### 3.4 Modifications and equivalences

In the supplement, we show that two modifications of the basic SF formulation also have equivalences in the regression perspective: namely, shrinking the empirical covariance in (13) towards the identity is equivalent to adding a ridge (squared $\ell_2$) penalty to the criterion in (14); and also, adding a null sensor at each state (one that always outputs 0) is equivalent to removing the constraints in (14). The latter equivalence here provides indirect but fairly compelling evidence that the constraints in the regression formulation (14) play an important role (under the model (2)): it says that removing them is equivalent to including meaningless null sensors, which intuitively should worsen its predictions.

## 4 Flu nowcasting application

**Experimental setup.** We examine the performance of our methods for nowcasting (one-week-ahead prediction of) wILI across 5 flu seasons, from 2013 to 2018 (total of 140 weeks). Recall the setup described in Section 3.2, with $k = 51$ states and $d = 308$ measurements. At week $t + 1$, we derive an estimate $\hat{x}_{t+1}$ of the current wILI in the 51 US states, based on sensors $z_{t+1}$ (each sensor being the output of an algorithm trained to predict wILI at a different geographic resolution from a given data source), and past wILI and sensor data. We consider 7 methods for computing the nowcast $\hat{x}_{t+1}$: (i) SF, or equivalently, constrained regression (14); (ii) SF as in (14), but with an additional ridge (squared $\ell_2$) penalty (equivalently, SF with covariance shrinkage); (iii) SF as in (14), but with an additional lasso ($\ell_1$) penalty; (iv/v) regression as in (14), but without constraints, and using a ridge/lasso penalty; (vi) random forests (RF) [Breiman, 2001], trained on all of the sensors; (vii) RF, but trained on all of the underlying data sources used to fit the sensors.

At prediction week $t + 1$, we use the last 3 years (weeks $t - 155$ through $t$) as the training set for all 7 methods. We do not implement unpenalized regression (as in (14), but without constraints), as it is not well-defined (156 observations and 308 covariates).[4] All ridge and lasso tuning parameters are chosen by optimizing one-week-ahead prediction error over the latest 10 weeks of data (akin to cross-validation, but for a time series context like ours). Python code for this nowcasting experiment is available at `http://github.com/mariajahja/kf-sf-flu-nowcasting`.

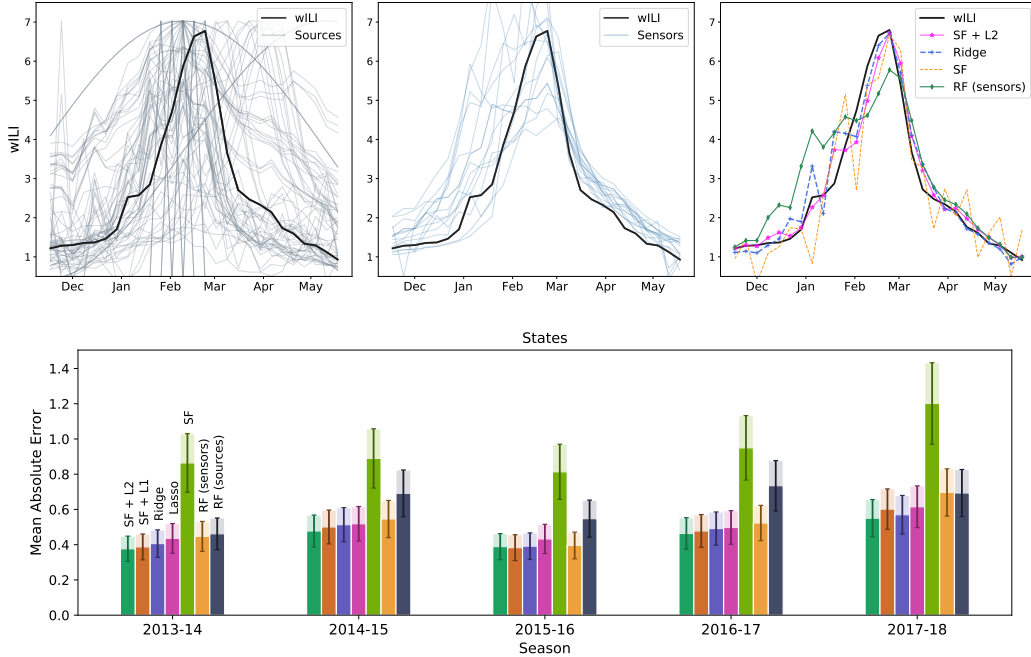

Figure 2: *Top row, from left to right: data sources, sensors, and nowcasts are compared to the underlying wILI values for Pennsylvania during flu season 2017-18. For visualization purposes, the sources are scaled to fit the range of wILI. On the rightmost plot, we display nowcasts using select methods. Bottom row: MAEs (full colors) and MADs (light colors) of nowcasts over 5 flu seasons from 2013-14 to 2017-18.*

**Missing data.** Unfortunately, sensors are observed at not only varying geographic resolutions, but also varying temporal resolutions (since their underlying data sources are), and missing values occur. In our experiments, we choose to compute predictions using the regression perspective, and apply a simple mean imputation approach (using only past sensor data), before fitting all models.

**Nowcasting results.** The bottom row of Figure 2 displays the mean absolute errors (MAEs) from one-week-ahead predictions by the 7 methods considered, averaged over the 51 US states, for each of the 5 seasons. Also displayed are the mean absolute deviations (MADs), in light colors. We see that SF with ridge regularization is generally the most accurate over the 5 seasons, SF with lasso regularization is a close second, and SF without any regularization is the worst. Thus, clearly, explicit regularization helps. Importantly, we also see that the constraints in the regression problem (14) (which come from its connection to SF) play a key role: in each season, SF with ridge regularization outperforms ridge regression, and SF with lasso regularization outperforms the lasso. Therefore, the constraints provide additional (beneficial) implicit regularization.

RF trained on sensors performs somewhat competitively. RF trained on sources is more variable (in some seasons, much worse than RF on sensors). This observation indicates that training the sensors is an important step for nowcasting accuracy, as this can be seen as a form of denoising, and suggests a view of all the methods we consider here (except RF on sources) as prediction assimilators (rather than data assimilators). Finally, the top row Figure 2 visualizes the nowcasts for Pennsylvania in the 2017-18 season. We can see that SF, RF (on sensors), and even ridge regression are noticeably more volatile than SF with ridge regularization.

## 5  Discussion and extensions

In this paper, we studied connections between the Kalman filter, sensor fusion, and regression. We derived equivalences between the first two and latter two, discussed the general implications of our results, and studied the application of our work to nowcasting the weekly influenza levels in the US. We conclude with some ideas for extending the constrained regression formulation (14) of SF.

**Sensor selection.** The problem of selecting a small number of relevant sensors (on which to perform sensor fusion) among a possibly large number, which we can call *sensor selection*, is quite a difficult problem. Beyond this, measurement selection in the Kalman filter is a generally difficult problem. As far as we know, this is an active and relatively open area of research. On the other hand, in regression, variable selection is extremely well-studied, and $\ell_1$ regularization (among many other tools) is now very well-developed (see, e.g., Hastie et al. [2009, 2015]). Starting from the regression formulation for SF in (14), it would be natural to add to the criterion an $\ell_1$ or *lasso* penalty [Tibshirani, 1996] to select relevant sensors,

$$\operatorname*{minimize}_{b_j \in \mathbb{R}^d} \quad \frac{1}{t}\sum_{i=1}^{t}(x_{ij} - b_j^T z_i)^2 + \lambda_j\|b_j\|_1$$
$$\text{subject to} \quad H^T b_j = e_j, \tag{16}$$

where $\|b_j\|_1 = \sum_{\ell=1}^{k}|b_{j\ell}|$, $j = 1, \dots, k$. It is not clear (nor likely) that (16) has an equivalent SF formulation, but the exact equivalence when $\lambda_j = 0$ suggests that (16) could be a reasonable tool for sensor selection. (Indeed, without even considering its sensor selection capabilities, this performed respectably for predictive purposes in the experiments in Section 4.) Further, we can perform a kind of process model selection with (16) by augmenting our measurement vector with multiple candidate process models, and penalizing only their coefficients. An example is given in the supplement.

**Joint sensor learning.** In the flu nowcasting problem, recall, the sensors are outputs of predictive models, each trained individually to predict wILI from a particular data source (flu proxy). Denote by $u_i \in \mathbb{R}^d$, $i = 1, \dots, t$ the data sources at times 1 through $t$. Instead of learning the sensors (predictive transformations of these sources) individually, we could learn them jointly, by extending (14) into:

$$\operatorname*{minimize}_{f_j \in \mathcal{F}_j} \quad \frac{1}{t}\sum_{i=1}^{t}\left(x_{ij} - b_j^T f_j(u_i)\right)^2 + \lambda_j P_j(f_j)$$
$$\text{subject to} \quad H^T b_j = e_j. \tag{17}$$

for $j = 1, \dots, k$. Here, each $\mathcal{F}_j$ is a space of functions from $\mathbb{R}^d$ to $\mathbb{R}^d$ (e.g., diagonal linear maps) and $P_j$ is a penalty to be specified by the modeler (e.g., the Frobenius norm in the linear map case). The key in (17) is that we are simultaneously learning the sensors and assimilating them.

**Gradient boosting.** Solving (17) is computationally difficult (even in the simple linear map case, it is nonconvex). An alternative that is more tractable is to proceed iteratively, in a manner inspired by *gradient boosting* [Friedman, 2001]. For each $j = 1, \dots, d$, let $A_j$ be an algorithm ("base learner") that we use to fit sensor $j$ from data source $j$. Write $y_i = Hx_i$, $i = 1, \dots, t$, and let $\eta > 0$ be a small fixed learning rate. To make a prediction at time $t + 1$, we initialize $x_i^{(0)} = 0$, $i = 1, \dots, t+1$ (or initialize at the fits from the usual linear SF), and repeat, for boosting iterations $b = 1, \dots, B$:

- For $j = 1, \dots, d$:
  - Let $y_{ij}^{(b-1)} = (Hx^{(b-1)})_{ij}$, for $i = 1, \dots, t$.
  - Run $A_j$ with responses $\{y_{ij} - y_{ij}^{(b-1)}\}_{i=1}^{t}$ and covariates $\{u_{ij}\}_{i=1}^{t}$, to produce $\bar{f}_j^{(b)}$.
  - Define intermediate sensors $z_{ij}^{(b)} = \bar{f}_j^{(b)}(u_{ij})$, for $i = 1, \dots, t+1$.
- For $j = 1, \dots, k$:
  - Run SF as in (14) (possibly with regularization) with responses $\{x_{ij} - x_{ij}^{(b-1)}\}_{i=1}^{t}$ and covariates $\{z^{(b)}\}_{i=1}^{t}$, to produce $\hat{b}_j$.
  - Define intermediate state fits $\bar{x}_{ij}^{(b)} = \hat{b}_j^T z_i^{(b)}$, for $i = 1, \dots, t+1$.
  - Update total state fits $x_{ij}^{(b)} = x_{ij}^{(b-1)} + \eta \bar{x}_{ij}^{(b)}$, for $i = 1, \dots, t+1$.

We return at the end our final prediction $\hat{x}_{t+1} = x_{t+1}^{(B)}$. It would be interesting to pursue this approach in detail, and study the extent to which it can improve on the usual linear SF.

**Acknowledgments.** We thank Logan Brooks for several helpful conversations and brainstorming sessions. MJ was supported by NSF Graduate Research Fellowship No. DGE-1745016. RR and RJT were supported by DTRA Contract No. HDTRA1-18-C-0008.

## Footnotes

[1] To make this unambiguous, we may take, say, $Q = aI$ and let $a \to \infty$.

[2] "Sensor fusion" is typically used as a generic term, similar to "data assimilation"; we use it to specifically describe the estimate in (8) to distinguish it from the KF. This is useful when we describe equivalences, shortly.

[3]This is more than just a hypothetical system; it is fully operational, and run by the Carnegie Mellon DELPHI group to provide real-time nowcasts of flu incidence every week, in all US states, plus select regions, cities, and territories. (See `https://delphi.midas.cs.cmu.edu`).

[4]SF is still well-defined, due of the constraint in (14): a nonunique solution only occurs when the (random) null space of the covariate matrix has a nontrivial intersection with the null space of $H^T$, which essentially never happens.

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
