[Supplementary Material · supp.pdf]

# Supplement to "Kalman Filter, Sensor Fusion, and Constrained Regression: Equivalences and Insights"

**Maria Jahja**
Department of Statistics
Carnegie Mellon University
Pittsburgh, PA 15213
maria@stat.cmu.edu

**David Farrow**
Computational Biology Department
Carnegie Mellon University
Pittsburgh, PA 15213
dfarrow0@gmail.com

**Roni Rosenfeld**
Machine Learning Department
Carnegie Mellon University
Pittsburgh, PA 15213
roni@cs.cmu.edu

**Ryan J. Tibshirani**
Department of Statistics
Machine Learning Department
Carnegie Mellon University
Pittsburgh, PA 15213
ryantibs@stat.cmu.edu

This document provides additional details, proofs, and simulation results for "Kalman Filter, Sensor Fusion, and Constrained Regression: Equivalences and Insights".

## A.1  Proof of Theorem 1

We can write the sensor fusion update as

$$\tilde{P}_{t+1} = (\tilde{H}^T \tilde{R}_{t+1}^{-1} \tilde{H})^{-1}$$
$$\hat{x}_{t+1} = \tilde{P}_{t+1} \tilde{H}^T \tilde{R}_{t+1}^{-1} \tilde{z}_{t+1},$$

where

$$\tilde{P}_{t+1} = (H^T R^{-1} H + \bar{P}_{t+1}^{-1})^{-1}.$$

By the Woodbury matrix identity, $(A + UCV^{-1}) = A^{-1} - A^{-1}U(C^{-1} + VA^{-1}U)^{-1}VA^{-1}$, with $A = \bar{P}_{t+1}^{-1}$ in our case, we get

$$\begin{aligned}
\tilde{P}_{t+1} &= \bar{P}_{t+1} - \bar{P}_{t+1}H^T(R + H\bar{P}_{t+1}H^T)^{-1}H\bar{P}_{t+1} \\
&= (I - \bar{P}_{t+1}H^T(R + H\bar{P}_{t+1}H^T)^{-1}H)\bar{P}_{t+1} \\
&= (I - K_{t+1}H)\bar{P}_{t+1},
\end{aligned} \quad (A.1)$$

where recall, the Kalman gain $K_{t+1}$ is defined in (6).

Now let us we rewrite the Kalman gain as

$$\begin{aligned}
K_{t+1} &= \bar{P}_{t+1}H^T(R + H\bar{P}_{t+1}H^T)^{-1} \\
&= \bar{P}_{t+1}H^T R^{-1}(I + H\bar{P}_{t+1}H^T R^{-1})^{-1},
\end{aligned}$$

so that

$$K_{t+1}(I + H\bar{P}_{t+1}H^T R^{-1}) = \bar{P}_{t+1}H^T R^{-1},$$

and after rearranging,

$$K_{t+1} = (I - K_{t+1}H)\bar{P}_{t+1}H^T R^{-1}. \quad (A.2)$$

Putting (A.1) and (A.2) together, we get

$$
\begin{aligned}
\tilde{P}_{t+1}\tilde{H}^T \tilde{R}_{t+1}^{-1}\tilde{z}_{t+1} &= (I - K_{t+1}H)\bar{P}_{t+1}(H^T R^{-1}z_{t+1} + \bar{P}_{t+1}^{-1}\bar{x}_{t+1}) \\
&= (I - K_{t+1}H)\bar{P}_{t+1}H^T R^{-1}z_{t+1} + (I - K_{t+1}H)\bar{x}_{t+1} \\
&= K_{t+1}z_{t+1} + (I - K_{t+1}H)\bar{x}_{t+1} \\
&= \bar{x}_{t+1} + K_{t+1}(z_{t+1} - H\bar{x}_{t+1}),
\end{aligned}
$$

which is exactly the Kalman filter prediction, completing the proof.

## A.2  Derivation of (12)

We first make the EKF estimate precise. Let

$$
\begin{aligned}
F_{t+1} &= Df(\hat{x}_t), & \text{(A.3)} \\
H_{t+1} &= Dh(\bar{x}_{t+1}), & \text{(A.4)}
\end{aligned}
$$

and define

$$
\begin{aligned}
\bar{x}_{t+1} &= F_{t+1}\hat{x}_t, & \text{(A.5)} \\
\hat{x}_{t+1} &= \bar{x}_{t+1} + K_{t+1}\big(z_{t+1} - h(\bar{x}_{t+1})\big), & \text{(A.6)}
\end{aligned}
$$

where $K_{t+1} \in \mathbb{R}^{k \times d}$ is defined via

$$
\begin{aligned}
\bar{P}_{t+1} &= F_{t+1}P_t F_{t+1}^T + Q, & \text{(A.7)} \\
K_{t+1} &= \bar{P}_{t+1}H_{t+1}^T(H_{t+1}\bar{P}_{t+1}H_{t+1}^T + R)^{-1}, & \text{(A.8)} \\
P_{t+1} &= (I - K_{t+1}H_{t+1})\bar{P}_{t+1}, & \text{(A.9)}
\end{aligned}
$$

Note that (A.7)–(A.9) are exactly the same as (5)–(7), with $F_{t+1}, H_{t+1}$ replacing $F, H$, respectively. Moreover, (A.5), (A.6) are *nearly* the same as (3), (4), with again $F_{t+1}, H_{t+1}$ replacing $F, H$, except that the residual in (A.6) is $z_{t+1} - h(\bar{x}_{t+1})$, and not $z_{t+1} - H_{t+1}\bar{x}_{t+1}$, as would be analogous from (4).

Next, we make what we called the extended SF (ESF) estimate precise. Let $\tilde{z}_{t+1} \in \mathbb{R}^{d+k}$ be as in (11), let $\tilde{H}_{t+1} \in \mathbb{R}^{(d+k)\times k}$ be the rowwise concatenation of $H_{t+1}$ and $I \in \mathbb{R}^{k \times k}$, and $\tilde{R}_{t+1}$ be as in (9). Here, $F_{t+1}, H_{t+1}, \bar{P}_{t+1}$ are as defined in (A.3), (A.4), (A.7), respectively. The ESF estimate is

$$
\hat{x}_{t+1} = (\tilde{H}^T \tilde{R}_{t+1}^{-1}\tilde{H})^{-1}\tilde{H}^T \tilde{R}_{t+1}^{-1}\tilde{z}_{t+1}. \tag{A.10}
$$

To see that (A.10) and (A.6) are equal, note that by following the proof of Theorem 1 directly, with $F_{t+1}, H_{t+1}$ in place of $F, H$, we get

$$
(\tilde{H}_{t+1}^T \tilde{R}_{t+1}^{-1}\tilde{H}_{t+1})^{-1}\tilde{H}_{t+1}^T \tilde{R}_{t+1}^{-1}\tilde{z}_{t+1} = \bar{x}_{t+1} + K_{t+1}(z_{t+1} - H_{t+1}\bar{x}_{t+1}).
$$

Adding and subtracting $K_{t+1}h(\bar{x}_{t+1})$ to the right-hand side gives

$$
\begin{aligned}
(\tilde{H}_{t+1}^T &\tilde{R}_{t+1}^{-1}\tilde{H}_{t+1})^{-1}\tilde{H}_{t+1}^T \tilde{R}_{t+1}^{-1}(z_{t+1}, \bar{x}_{t+1}) \\
&= \bar{x}_{t+1} + K_{t+1}\big(z_{t+1} - h(\bar{x}_{t+1})\big) + K_{t+1}(h(\bar{x}_{t+1} - H_{t+1}\bar{x}_{t+1}) \\
&= \bar{x}_{t+1} + K_{t+1}\big(z_{t+1} - h(\bar{x}_{t+1})\big) + (I - K_{t+1}H_{t+1})\bar{P}_{t+1}H_{t+1}^T R^{-1}(h(\bar{x}_{t+1} - H_{t+1}\bar{x}_{t+1}) \\
&= \bar{x}_{t+1} + K_{t+1}\big(z_{t+1} - h(\bar{x}_{t+1})\big) + \tilde{P}_{t+1}H_{t+1}^T R^{-1}(h(\bar{x}_{t+1} - H_{t+1}\bar{x}_{t+1}),
\end{aligned}
$$

where in the second line we used (A.2), and in the third we used (A.1). Rearranging gives

$$
(\tilde{H}_{t+1}^T \tilde{R}_{t+1}^{-1}\tilde{H}_{t+1})^{-1}\tilde{H}_{t+1}^T \tilde{R}_{t+1}^{-1}\big(z_{t+1}+H_{t+1}\bar{x}_{t+1}-h(\bar{x}_{t+1}),\ \bar{x}_{t+1}\big) = \bar{x}_{t+1}+K_{t+1}\big(z_{t+1}-h(\bar{x}_{t+1})\big),
$$

which is precisely the desired conclusion, in (12).

## A.3 Proof of Theorem 2

Let us denote $X \in \mathbb{R}^{t \times k}$ and $Z \in \mathbb{R}^{t \times d}$ the matrices of states and sensors, respectively, for the first $t$ time points. That is, $X$ has rows $x_i \in \mathbb{R}^k$, $i = 1, \ldots, t$ and $Z$ has rows $z_i \in \mathbb{R}^d$, $i = 1, \ldots, t$. Fix any $j = 1, \ldots, k$. Let $\hat{a}_j \in \mathbb{R}^d$ be the $j$th column of $\hat{R}_{t+1}^{-1} H (H^T \hat{R}_{t+1}^{-1} H)^{-1}$, and let $\hat{b}_j \in \mathbb{R}^d$ be the solution of (14), equivalently, the solution of

$$\begin{aligned} \underset{b_j \in \mathbb{R}^d}{\text{minimize}} \quad & \|X_j - Z b_j\|_2^2 \\ \text{subject to} \quad & H^T b_j = e_j, \end{aligned} \tag{A.11}$$

where $X_j$ denotes the $j$th column of $X$. We will show that $\hat{a}_j = \hat{b}_j$.

The Lagrangian of problem (A.11) is

$$L(b_j, u_j) = \|X_j - Z b_j\|_2^2 + u_j^T (H^T b_j - e_j),$$

for a dual variable (Lagrange multiplier) $u_j \in \mathbb{R}^k$. Taking the gradient of the Lagrangian and setting it equal to zero at an optimal pair $(\hat{b}_j, \hat{u}_j)$ gives

$$0 = Z^T (Z \hat{b}_j - X_j) + H \hat{u}_j,$$

and rearranging gives

$$\hat{b}_j = (Z^T Z)^{-1} (Z^T X_j - H \hat{u}_j). \tag{A.12}$$

The dual solution $\hat{u}_j$ can be determined by plugging (A.12) into the equality constraint $H^T \hat{b}_j = e_j$, but for our purposes, the explicit dual solution is unimportant.

We will now show that $\hat{b}_j = \hat{R}_{t+1}^{-1} H \hat{\beta}_j$ for some $\hat{\beta}_j \in \mathbb{R}^k$. Write

$$\begin{aligned} \hat{R}_{t+1} &= \frac{1}{t} (Z - X H^T)^T (Z - X H^T) + (1 - \alpha) I \\ &= \frac{1}{t} (Z^T Z - H X^T Z - Z^T X H^T + H X^T X H^T). \end{aligned}$$

Then

$$\begin{aligned} \hat{R}_{t+1} \hat{b}_j &= \frac{1}{t} (Z^T Z \hat{b}_j - H X^T Z \hat{b}_j - Z^T X H^T \hat{b}_j + H X^T X H^T \hat{b}_j) \\ &= \frac{1}{t} (Z^T X_j - H \hat{u}_j - H X^T Z \hat{b}_j - Z^T X_j + H X^T X_j) \\ &= H \underbrace{\left( \frac{X^T X_j - \hat{u}_j - X^T Z \hat{b}_j}{t} \right)}_{\hat{\beta}_j}, \end{aligned}$$

as desired, where in the second line we have used (A.12) and the constraint $H^T \hat{b}_j = e_j$.

Observe that $\hat{a}_j = \hat{R}_{t+1}^{-1} H \hat{\alpha}_j$ for some $\hat{\alpha}_j \in \mathbb{R}^k$, in particular, for $\hat{\alpha}_j$ defined to be the $j$th column of $(H^T \hat{R}_{t+1}^{-1} H)^{-1}$. Further,

$$e_j = H^T \hat{a}_j = H^T \hat{b}_j$$

the constraint on $\hat{a}_j$ holding by direct verification, and the constraint on $\hat{b}_j$ holding by construction in (A.11). That is,

$$H^T \hat{R}_{t+1}^{-1} H \hat{\alpha}_j = H^T \hat{R}_{t+1}^{-1} H \hat{\beta}_j,$$

and since $H^T \hat{R}_{t+1}^{-1} H$ is invertible, this leads to $\hat{\alpha}_j = \hat{\beta}_j$, and finally $\hat{a}_j = \hat{b}_j$, completing the proof.

## A.4 Further SF-regression equivalences

### A.4.1 More regularization: covariance shrinkage

Covariance shrinkage—which broadly refers to the technique of adding a well-conditioned matrix to a covariance estimate to provide stability and regularity—is widely used and well-studied in modern

multivariate statistics, data mining, and machine learning. As such, it would be natural to replace the empirical covariance matrix estimate (13) for the measurement noise covariance by

$$\hat{R}_{t+1} = \frac{\alpha}{t} \sum_{i=1}^{t} (z_i - Hx_i)(z_i - Hx_i)^T + (1-\alpha)I, \tag{A.13}$$

for a parameter $\alpha \in [0, 1]$. For sensor fusion in the flu nowcasting problem, this is considered (in some form) in Farrow [2016], and leads to significant improvements in nowcasting accuracy.

Our next result shows that when we use shrinkage as in (A.13) to estimate the measurement noise covariance in SF, this is equivalent to adding a ridge penalty in the regression formulation.

**Corollary 1.** *Let $\hat{R}_{t+1}$ be as in (A.13), for some value $\alpha \in [0, 1]$. Consider the SF prediction at time $t + 1$, with $\hat{R}_{t+1}$ in place of R, denoted $\hat{x}_{t+1} = \hat{B}^T z_{t+1}$. Then each column of $\hat{B}$, denoted $\hat{b}_j \in \mathbb{R}^d$, $j = 1, \ldots, k$, solves*

$$\begin{aligned}
\underset{b_j \in \mathbb{R}^d}{\text{minimize}} \quad & \frac{1}{t} \sum_{i=1}^{t} (x_{ij} - b_j^T z_i)^2 + \frac{(1-\alpha)}{\alpha} \|b_j\|_2^2 \\
\text{subject to} \quad & H^T b_j = e_j.
\end{aligned}$$

*Proof.* As before, let $X \in \mathbb{R}^{t \times k}$ and $Z \in \mathbb{R}^{t \times d}$ denote the matrix of states and sensors, respectively, over the first $t$ time points. We can write $\hat{R}_{t+1}$ in (A.13)

$$\frac{\alpha}{t}(Z - XH^T)^T(Z - XH^T) + (1-\alpha)I = \frac{1}{t}(\tilde{Z} - \tilde{X}H^T)^T(\tilde{Z} - \tilde{X}H^T),$$

where $\tilde{Z} \in \mathbb{R}^{(t+d) \times d}$ is the rowwise concatenation of $\sqrt{\alpha/t}Z$ and $\sqrt{1-\alpha/t}I$, and $\tilde{X} \in \mathbb{R}^{(t+k) \times k}$ is the rowwise concatenation of $\sqrt{\alpha/t}X$ and $0 \in \mathbb{R}^{k \times k}$ (the matrix of all 0s). Applying Theorem 2 to $\tilde{X}, \tilde{Z}$, expanding the criterion in the regression problem, and then multiplying the criterion by $1/\alpha$, gives the result. $\qquad \square$

### A.4.2 Less regularization: zero padding

In the opposite direction, we now show that we can modify SF and obtain an equivalent regression formulation with less regularization, specifically, without constraints.

**Corollary 2.** *At each $t = 1, 2, 3, \ldots$, suppose we augment our measurement vector by introducing $k$ measurements that are identically zero, denoted $\tilde{z}_t = (z_t, 0) \in \mathbb{R}^{d+k}$. Suppose that we augment our measurement map accordingly, defining $\tilde{H} \in \mathbb{R}^{(d+k) \times k}$ to be the rowwise concatention of H and the identity $I \in \mathbb{R}^{k \times k}$. Consider running SF on this augmented system, using the empirical covariance to estimate R, and let $\hat{x}_{t+1} = \hat{B}^T z_{t+1}$ denote the SF prediction at time $t + 1$. Then each column of $\hat{B}$, denoted $\hat{b}_j \in \mathbb{R}^d$, $j = 1, \ldots, k$, solves*

$$\underset{b_j \in \mathbb{R}^d}{\text{minimize}} \sum_{i=1}^{t} (x_{ij} - b_j^T z_i)^2.$$

*Proof.* Applying Theorem 2 to the augmented system gives the equivalent regression problem

$$\begin{aligned}
\underset{b_j \in \mathbb{R}^d, a_j \in \mathbb{R}^k}{\text{minimize}} \quad & \sum_{i=1}^{t} (x_{ij} - b_j^T z_i - a_j^T 0)^2 \\
\text{subject to} \quad & H^T b_j + a_j = e_j.
\end{aligned}$$

The constraint is satisfied with $a_j = e_j - H^T b_j$. But $a_j$ has no effect on the criterion, so the constraint can be removed. $\qquad \square$

**Remark A.1.** The analogous equivalence holds for covariance shrinkage and ridge regression. That is, in Corollary 2, if instead of the empirical covariance, we use $\alpha$ times the empirical covariance plus $(1-\alpha)I$, then SF on the augmented system is equivalent to unconstrained ridge, at tuning parameter $(1-\alpha)/\alpha$.

## A.5   Example of process model selection

Here we give a simple empirical example of process model selection using the regression formulation of SF. We initialized $x_0 = 1$, and generated data according to

$$x_t = 0.5x_{t-1} + 0.05\sin(0.126t) + \delta_t,$$
$$z_t = Hx_t + \epsilon_t,$$

for $t = 1, \ldots, 200$. Here $H \in \mathbb{R}^{4 \times 1}$ is simply the column vector of all 1s, and the noise is drawn as $\delta_t \sim N(0, 0.01)$, $\epsilon_t \sim N(0, I)$, independently, over $t = 1, \ldots, 150$.

The prediction setup is as follows. At each time $t + 1$, when making a prediction of $x_{t+1}$, we observe all past states $x_i$, $i = 1, \ldots, t$ and all measurements $z_i$, $i = 1, \ldots, t + 1$. We fit 5 different candidate process models to past state data:

1. linear autoregression;
2. quadratic autoregression;
3. spline regression on time;
4. sine regression on time;
5. cosine regression on time.

To be clear, models 1 and 2 regress $x_i$ on $x_{i-1}$ and $x_{i-1}^2$, respectively, over $i = 1, \ldots, t$. Models 3–5 regress $x_i$ on a spline, sine, and cosine transformation of $i$, respectively, over $i = 1, \ldots, t$. The sine and cosine transformations are given the true frequency. The spline is a cubic smoothing spline (with a knot at every data point) and its tuning parameter is chosen by cross-validation (using only the past data). After being fit, we use each of the candidate process models 1–5 to make a prediction of $x_{t+1}$, given $z_{t+1}$. We take this as its ouput.

For $t = 151, \ldots, 200$, we define $\tilde{z}_t \in \mathbb{R}^9$ to be the measurement vector $z_t \in \mathbb{R}^4$ augmented with the outputs of the 5 candidate process models as described above (the burn-in period of 150 time points ensures that the candidate process models have enough training data to make reasonable predictions). Figure A.1 shows the outputs from these models over the last 50 time points.

Figure A.1: *Simple process model selection example: outputs from 5 candidate process models, over the last 50 time points.*

Finally, in the last 50 time points, to get an assimilated prediction of $\hat{x}_{t+1}$ at each time $t + 1$, we solve the constrained regression problem with a lasso penalty (16), using cross-validation to select $\lambda$ (again, using only past data). Further, we penalize only the coefficients of the candidate process

models (not the pure measurements). Table 1 shows the median of the coefficients over the last 50 time points (in this table, the coefficients for the pure measurement sensors are aggregated as one). We see that the lasso tends to select the linear and sine sensors, as expected (because these two make up the true dynamical model), and places a small weight on the spline sensor (which is flexible, and can mimic the contribution of the sine sensor).

| | Linear | Quadratic | Spline | Sine | Cosine | Measurements |
|---|---|---|---|---|---|---|
| Median Coefficient | 0.643 | 0.000 | 0.094 | 0.189 | 0.000 | 0.0175 |

Table 1: *Simple process model selection example: median regression coefficients for the sensors, over the last 50 time points.*

# References

David Farrow. *Modeling the Past, Present, and Future of Influenza*. PhD thesis, Computational Biology Department, Carnegie Mellon University, 2016.