[Reviews · NeurIPS 2019]

Reviewer 1



The paper considers an elegant new perspective on Kalman Filters, by demonstrating how it might be considered a sensor fusion problem by absorbing the process error as a block diagonal term of the sensor error covariance matrix. Further, the paper builds on this to formulate time-series data prediction from sensors as a regression based on empirical covariance estimates of the sensor error. This tool is then used to formulate prediction of flu activity from proxy measurements. The formulation is well motivated and the results are good. The supplement offers further explanation of how covariance shrinkage might be viewed as being within this framework. This particular version yields good results in the use case demonstrated. Further discussions on how the new formulation can lead to sensor selection (based on significance) are useful, but preliminary. A minor nitpick is that the authors speculate that non-linear models could have offered further benefits, but do not complete the evaluation.

Reviewer 2



Rebuttal acknowledged, thank you for the additional clarifications. --- Originality: I believe that the findings of Section 2 are well-known given a Bayesian / Gaussian viewpoint of the KF (c.f. [1] and [2]). Indeed, given a flat prior for $x_{t+1}$ (i.e., Gaussian with "infinite" variance), we have two independent observations: - the influence of the past (prediction term) - the influence of the current measurement (filtering term) both have Gaussian likelihood. So the posterior density of $x_{t+1}$ is proportional to a product of three Gaussian-shaped terms. The two different ways in which these terms can be folded into each other (using standard Gaussian conjugacy rules) lead to Thm 1. I believe that the linear-algebraic formulation the authors use just hides the fact that we are multiplying Gaussian PDFs in different ways. On the other hand, I think that the reformulation of Section 3 is less straightforward and perhaps of larger interest. Just like the connection between the KF and linear-Gaussian models opened up many new possibilities, I believe the authors' reformulation may lead to practical improvements, such the ones outlined in Section 5. Quality: Generally speaking, the paper is sound and rigorous. The experimental evaluation of Section 4 is well explained. The results are somewhat underwhelming: it is not clear that the proposed method does significantly better than competing ones. I believe that the paper would benefit from having some of the new ideas outlined in Section 5 explored in the flu-nowcasting applications. E.g., maybe adding L1-regularization helps? Clarity: Generally speaking, the paper is well-written and easy to follow. The supplementary material is also clear and well-structured. Table 1 is not easy to parse. I would suggest presenting the results in a bar plot instead. - line 67: "no a process" -> typo - line 87: "intuitive extend"' -> typo [1]: Faragher, Understanding the basis of the Kalman filter via a simple and intuitive derivation, 2012 [2] Särkkä, S. Bayesian Filtering and Smoothing, 2013

Reviewer 3



I am afraid that I struggle to see what is new in this paper and what its significance is. As stated by the authors themselves the main result in Theorem 1 “is elementary”, like an exercise for a linear systems course. There has been quite a lot of work on viewing the Kalman filtering problem as a convex optimisation problem, see e.g. https://web.stanford.edu/~boyd/papers/pdf/rt_cvx_sig_proc.pdf I have a feeling that this view will help the authors a fair bit in developing their ideas further and find relationships with more existing work. A detail: I would recommend the authors to reconsider the extremely narrow definition you are using for the word sensor fusion. Sensor fusion is a much wider term, see e.g. its Wikipedia entry https://en.wikipedia.org/wiki/Sensor_fusion Academically a lot of research related to sensor fusion is published at the yearly fusion conference https://fusion2019.org/ From just a very quick look at these sites it should be clear that sensor fusion is much broader that the particular linear equation (8) in the current manuscript.

[Author Response · NeurIPS 2019]

**Reviewer #1**

Thank you for your encouraging comments. We are glad that you are able to recognize and understand the main contributions in our paper.

We have more to say about nonlinear models (including further experiments), which we can add to the camera-ready version. Basically, a version of the constraints in the regression formulation of sensor fusion can be also imposed for nonlinear models. We suggested this at the end of our paper in equation (17), but have since discovered a more tractable way to write the constraints in terms of local linear approximations.

**Reviewer #2**

Thank you for your thorough and helpful review. We appreciate all of your feedback. Bayesian viewpoint: this is a fair point, and we essentially agree with everything you say here. While we did not find something like Theorem 1 stated explicitly in the Faragher paper, nor in the Sarkka book (however, it is possible we missed it and we will look carefully through this entire book, before making this claim in any official way), we agree that the Bayesian/Gaussian viewpoint offer another lens from which we can understand Theorem 1. We are happy to add this along with appropriate references and discussion in a camera-ready version of our paper.

Thus by itself, Theorem 1 is not of high originality (for the reasons just noted, and to repeat, we will revise the text surrounding it appropriately). But we still think its role in our paper is significant, and hence it should be kept in the paper. This is because Theorem 1, combined with the insight in Theorem 2 that we can reformulate sensor fusion via a *forwards* or *direct* regression, suggests new and promising methodological possibilities. For example, we can throw in multiple candidate process models as sensors, then perform feature selection in the regression formulation to adaptively select some subset of them. This is explained in the "sensor selection" paragraph at the end of the paper and demonstrated empirically in Section A.7 in the supplement.

We are glad that you understand and appreciate the significance of Theorem 2. Empirical results/better demonstrations on real-world data: we have since rerun our experiments in Table 1 at the *US state level*, over more seasons. (This gives a test set with over 50x more observations: Table 1 only reports results at the national level over 4 seasons, which is much sparser in terms of a test set evaluation.) Sensor fusion combined with shrinkage now consistently displays a clear advantage over random forests (and all others, though random forests its closest competitor) throughout. This is important because it shows that even just a simple linear model with the "right" hierarhical constraints, encoding the measurement map, can perform well in comparison to a much richer, nonlinear/nonparametric method like random forests. It also suggests that with the "right" constraints put in place, a nonlinear method should do very well.

We are happy to add another sensor selection experiment to the supplement in a camera-ready version, rerunning an experiment like that in Section A.7 on real data. For example, we can try multiple process models on the flu data.

**Reviewer #3**

Thank you for your feedback; we regret that you were not able to appreciate our contributions and we will revise the paper accordingly to try to make the main points more salient. We hope that in the meantime, our response here will help clarify some things. First, we agree that Theorem 1 is simple and in it of itself of major originality. But we still believe that its implications when combined with Theorem 2 are significant, and believe that as such it belongs in our paper. Please see our comments in response to a similar point raised by Reviewer #2, above.

The name "sensor fusion": we are aware that this is a very broad term and describes a whole class of methods, not just equation (8). We apologize for the confusion. We simply needed something to call (8) in order to cleanly refer to it in our paper. See the bottom footnote on page 2 of our paper. We can definitely change this in a camera-ready version of the paper and are happy to hear alternative suggestions for a name. But to be clear, the fact that "sensor fusion" is a broad term (and has conferences associated with it) should not take anything away from the equivalences derived in our paper. We could have simply called this something else ("process-agnostic Kalman filter", or "linear inverse MLE").

Convex optimization view of Kalman filtering: we appreciate you raising this point, and sending an example reference, but we are in fact quite familiar with the optimization view of many estimation/tracking/control/prediction tasks, including Kalman filtering (the Boyd and Vandenberghe book, for example, makes this view ubiquituous). We ourselves work actively in this area as well. However, we must be perfectly clear that *this is not the same as the equivalence* as that we derive in Theorem 2, and the convex optimization view of KF has no bearing on the originality nor significance of this result. The work by Boyd and others just poses planning as a convex optimization problem, which is quite natural (and apparently, effective). Our Theorem 2 is completely different. It reformulates a *backwards* or *indirect* model of $z_{t+1}|x_{t+1}$ in terms of a *forwards* or *direct* model of predicting $x_{t+1}$ from $z_{t+1}$. This has significant implications because it allows us to bring in the entire "ML toolbox" for this prediction problem. We can expand on the simulation in Section A.7 of the supplement, if you think this would help (see our comments to Reviewer #2 about this, too).

[Meta-Review · NeurIPS 2019]

The results of this paper could be of interest to NeurIPS. However, the author(s) should try and address most of the concerns raised by the reviewers, specially a review of existing work on Kalman filtering with infinite variance.